# Environmental Criteria for Assessing the Competitiveness of Public Tenders with the Replacement of Large-Scale LEDs in the Outdoor Lighting of Cities as a Key Element for Sustainable Development: Case Study Applied with PROMETHEE Methodology

**Manuel Jesús Hermoso-Orzáez** [1,*] **, José Adolfo Lozano-Miralles** [1] **, Rafael Lopez-Garcia** [2] **and Paulo Brito** [3]

1   Department of Graphic Engineering, Design and Projects, Universidad de Jaen, 23071 Jaen, Spain; jalm0017@red.ujaen.es
2   Department of Mechanical and Mining Engineering, Universidad de Jaén, 23071 Jaen, Spain; rlgarcia@ujaen.es
3   Instituto Politecnico, I.P.P Portalegre, 7300-110 Portalegre, Portugal; pbrito@ipportalegre.pt
*   Correspondence: mhorzaez@ujaen.es; Tel.: +34-610-389-020

**Abstract:** The technological change to LEDs is an unstoppable reality which, little by little, is becoming increasingly important in terms of the lighting inside and outside our homes. The exterior lighting of our cities is moving decisively and clearly towards the incorporation of this technology in urban spaces. The energy efficiency, light quality, and economic benefits of LED technology are an unquestionable reality. This is causing public administration projects involving large-scale switches to LEDs to be promoted and financed; however, it is beginning to be observed that the commitment to the policies decided by this technology should take into account some environmental aspects which have not been studied to date. The environmental impact of the substitutions is caused by the need to valorize the replaced luminaires. Until now, most have been stored without the possibility of use, reuse, or recovery. The environmental impact produced in the manufacture of LED luminaires that replace the old sodium vapor (VSAP) or metal halide (MH) discharge lamps must also be considered. In addition, in the administrative clauses specifications that govern the public tenders, it is observed that the fundamental environmental aspects both of recycling the old lamps, and of the life cycle analysis (LCA) of the luminaires that are replacing them, have not been contemplated or valued with sufficient weight. In addition, there are very few public substitution contests in which environmental criteria are rewarded or valued in an important way. This work intends to summarize a methodological proposal using the techniques of multiple decision-making criteria for the selection of bidding companies for public outdoor lighting competitions. We propose the use of the PROMETHEE method multi-criteria analysis for the application of the most commonly used criteria for the luminaire LED selection process, including an environmental impact assessment with LCA techniques, and propose this as a case or model guide in the public contests of cities. A model of the bidding conditions that addresses and assesses the environmental aspects which are absolutely key to sustainable development is supported by the ecological criteria of the circular economy.

**Keywords:** Life cycle assessment LCA; LED luminaires; lighting public tender; streetlight waste; recycling waste; environmental criteria; decision tool; PROMETHEE; lighting pollution; circular economy

## 1. Introduction

Today, artificial lighting demands 20% of total global electricity production [1]. Its impact represents 1.9 billion tons of $CO_2$ emissions per year, which represents 70% of vehicle emissions worldwide [2]. Energy consumption in public lighting accounts for 2.3% of the world's electricity consumption [3] and, according to different studies, in the municipalities of developed countries, values of between 40% [4] and 60% of municipal electricity consumption can be reached [5]. A large number of public lighting installations were created between 30 and 40 years ago and are therefore obsolete [6]. Today, LED and/or OLED technology has reached high levels of efficiency (more than 276 lm/W) and increasingly low costs. In addition, its useful life is several times greater than that of discharge lamps [3]. Street lighting is an important part of transport infrastructure and public utilities. In Europe (EU28), there are more than 1.6 million km of illuminated streets that consume annually approximately 35 TW h with a cost of €4 billion for public authorities [6].

In the European Union (hereafter the EU), energy savings and efficiency policies aimed at urban lighting are being promoted [7]. Lighting installations in Spain total 8,849,839 points of light, which, with an average power of 156 W, represents an electricity consumption of 5296 GWh/year for the whole of Spain [8]. The new energy-saving policies promoted by the EU recommend the replacement of discharge lamps—sodium vapor (VSAP) or metal halide (MH)—with LED luminaires. Replacing the discharge lamps with LEDs with similar lighting performance allows a reduction of electricity consumption between 20% and 50%. [9–12]. Currently, there are a large number of reasons to upgrade street lighting—one reason is that LEDs are the most efficient light source, even more than high-pressure sodium (HPS) lamps [13,14]. LED luminaires and the associated equipment are considered to be technologically mature—a fact that considerably reduces their cost [15,16].

Numerous studies have analyzed the energy efficiency of these new luminaries by comparing them with fluorescent lamps [17] and MH lamps [18,19]. The reduction of LED prices, together with the high capacity of control, regulation, and remote management, allow us to obtain periods of the amortization of investments with terms of less than three years [20,21]. Studies conducted in the city of Detroit have allowed for the calculation of "simple payback" return periods with amortizations in less than 2.5 years [22]. In this global scenario, different companies and studies have generated reports that ensure that it is expected that, in developed countries, more than $53.7 million can be invested in LED street lighting during the period 2015–2025 [23].

However, the replacement of LED discharge lamps is encountering problems that were not common in lighting installations. Different studies have focused on comparatively evaluating both technologies, focusing on measuring the indexes of annoying glare, uniformity, possible contamination of the power grid, harmonic contamination, and trying to measure the dreaded cold peak currents in the ignition starts [24], as well as anomaly factors such as power factor (PF) and current total harmonic distortion (THD) [25]. At the time of the initial cold start of the installation, the "input peak currents" [26], and consequently the over-excitation of the diodes used by the LED sources (due to the voltage required during the start-up period), can cause overcurrent problems and unwanted disconnections [21,27–29].

The construction of buildings and roads—and in particular lighting installations—are responsible for the consumption of almost half of the raw materials and energy of the planet [30]. Consequently, the manufacture of products for construction in general and facilities in particular, such as outdoor lighting, has a great impact on the depletion of finite resources, in addition to the production of greenhouse gas emissions from the combustion of fossil fuels that are involved in equipment manufacturing processes. To reduce the associated greenhouse gas emissions and the resulting impact on the climate, it is necessary to use construction materials which are environmentally sustainable [31,32].

Current environmental sustainability policies and associated concepts of bioclimatic architecture, as well as social concern for general environmental aspects (global warming, greater damage to the ozone layer and the accumulation of waste), have made the construction industry increasingly sensitive

to the consideration of environmentally sustainable products and equipment. Materials that reduce energy consumption require the creation of innovative products that are sustainable. In fact, in Europe, the construction sector is responsible for 40–45% of primary energy consumption, which comprises a significant proportion of greenhouse gas emissions [33]. The use of sustainable materials in lighting equipment would help reduce these gas emissions.

With such an expectation, many studies are being developed that apply the LCA (Life Cycle Assessment) methodology to analyze the environmental impacts caused in the manufacture of equipment. A product manufactured with the criteria of a low environmental impact can cause the reduction of greenhouse gases by up to 75%, the reduction of production costs by 12% and an improvement of 2–5 times in the energy efficiency of production. In addition, the LCA model is currently being applied in numerous studies, such as one from Tsinghua University that aims to calculate the life cycle fossil energy consumption and greenhouse gas emissions in China. These studies show that it is necessary to evaluate the environmental impact of building materials for installations using the LCA technique. Many scientific studies that use the LCA methodology compare different materials, highlighting those with a minor impact on the environment [34,35].

On the other hand, environmental criteria are considered a key factor when selecting the most suitable LED equipment and luminaires to deal with mass substitutions. The life cycle analysis of the LED luminaire is becoming a priority aspect when selecting an LED luminaire to be replaced. Environmental analysis is becoming more frequent in terms of the different levels of impact that substitutions can cause in the medium and long term. It is for this reason that we intend to present a model of the technical specification in this work that allows us to incorporate, as criteria of assessment, the environmental impact criteria at the time of the massive replacement of discharge lamps with LED luminaires. A new technical report that is being prepared for green public procurement criteria presents the selection criteria that take into account the negative effects of environmental impacts [36,37].

LED lamps are still experiencing improvements in terms of their efficiency and design of materials, which raises questions about whether a longer life is desirable from a general environmental perspective. Applying a comprehensive evaluation of the LCA life cycle, actual product cases have been studied from 2012 to 2017—the research is based on previous studies of product life and lighting product research in order to determine the scenarios in which longer lives are desirable from a general environmental perspective. The factors explored in the scenarios include the improvement of products in terms of eco-efficiency, as well as the contexts of use of clean electricity. The results indicate that the replacement of a product with improved products resulted in environmental benefits compared to the use of longer-lasting products, but there is some compensation between environmental impacts. However, these compensations are minimized in the context of decarbonized electricity mixtures and will decrease further as the technology of the LED lamp matures technologically, and product development decreases the environmental impact of the manufacture of LED luminaires [38].

On the other hand, it seems that the multi-criteria decision tools [39,40] are becoming increasingly valuable in street lighting, especially for energy planning. Carli et al. [41,42] developed a multi-criteria decision-making tool for the support of public decision makers. This tool can help to select the optimum retrofit of an existing street lighting system in an urban area. The scope was to reduce energy consumption and environmental impacts, as well as maintaining the required comfort and the quality of life. This work aims to provide a model of technical conditions that will serve as a basis in public tenders with large-scale replacement by LED luminaires. Further, we aim to allow the contest evaluator of public tenders to use multicriteria decision tools such as PROMETHEE [43] to facilitate the choice of an LED luminaire that best suits the technical, light quality, savings and, of course, environmental sustainability requirements [37].

## 2. Methodology

At this point, we will define the objectives, the methodology pursued in this work and the case study that will serve as a methodological proposal.

*2.1. Objetive*

The promulgation of the energy efficiency regulations (hereafter REEIAE) in Spain together with the growth of LED technology [44] and the admission of the contracting of energy services companies (Energetics Services Companies) (ESC) by the public administrations are milestones that have emerged in recent years—the facilities in the EU and their method of management, maintenance and control are changing. The pilot experiences carried out by the IDAE (Institute for Diversification and Energy Saving) in the Town Halls of Alcorcón and Soto del Real (Spain) to adapt their outdoor lighting installations to the requirements of the REEIAE [44] through (ESC) and the lines of financing implemented with the aid programs of the National Energy Efficiency Fund and JESSICA FIDAE have revealed the high potential for savings in electricity consumption of this type of facility. In addition, this potential for savings in economic terms allows, in most cases, investments to be made with a simple return period of less than six years, which is ideal for the business of ESC companies [45].

The municipalities in the case of Spain propose the reform of a set of more than 700,000 existing light points, and more than 97% of the cases have opted for the replacement of discharge lamps with LED technology. With this, an average saving of 65% per year in electricity consumption has been achieved due to the variation in the power of the new light points, which drops from an average of 164 to 58 W per luminaire. Most of the government, especially the local administrations, are opting for public tenders for the concession of the lighting services for to a minimum of 10 years, including maintenance, management, and investments in the ESC (Services Energetics Business) modality [46].

However, in some municipalities, perhaps due to the rush generated by carrying out the substitutions in a precipitous manner, regardless of demanding minimum qualities [47] and due to the lack of a detailed study of the facilities or the qualities of the equipment to be replaced, unforeseen situations have arisen, such as massive blackouts associated with surges caused by thunderstorms, problems in the cold ignition of the lighting installation, or distortions in the network generated by the harmonic contamination associated with the electronic equipment of the LED drivers [21].

The EU Circular Economy Action Plan promotes a longer lifespan of the products to be used as building materials or in facilities; in particular, outdoor lighting. However, encouraging longer lives could also generate compensation between different environmental impacts for some product categories such as LED luminaires [3,4]. This is why the environmental impact analysis in the different categories obtained in the industrial manufacture of LED luminaires using LCA techniques has become a key and priority aspect to be taken into consideration in the preparation of the technical conditions to address the public bidding of projects, whose purpose is to address the technological changes to LEDs in external lighting, based on efficiency criteria as well as energy, light, economic and, of course, environmental requirements.

This work aims to propose an improved model of technical specifications, complementary to those proposed by the IDAE or the CIE (Spanish Lighting Committee) [45,46], which allows a municipal technician to guarantee that the replacement of LED luminaires will be carried out, with the maximum guarantees of technical performance, light energy, and a low environmental impact, allowing a municipal technician to face this change and its consequences in the medium and long term with the necessary peace of mind that at least the replaced product meets the standards of quality and sufficient light efficiency and respect for the environment. Finally, this specification model is discussed and analyzed, supported by studies and work carried out by specialists in the field.

A Specification of Technical Conditions type or model is proposed as a case study that will allow the replacement of 1034 luminaires with 100 W HM discharge lamps, with LED luminaires with low environmental impact, greater energy efficiency, and similar lighting performance. Contemplating, as a novelty, the analysis of environmental sensitivity of the LED luminaire using LCA techniques, it is intended that this improved technical specification model with respect to the one defined by the IDEA or the CIE will allows us to consider and improve some technical aspects that we consider determinants to improve the functionality and environmental sustainability of the lighting installation to be replaced.

### 2.2. Model Specification of Technical Conditions for Case Study and Valuation Criteria Including Environmental Impact Aspects

Next, we propose a model of the technical specifications that can serve as a reference in tenders for projects promoted by the public tenders, regardless of the variation of the weights of the different criteria, which can be carried out in line the best criteria by the bidder. In this case study, we intend to set some general guidelines to assess different criteria, and specifically the environmental criteria, as something innovative and necessary and which is widely demanded by a society which is becoming increasingly sensitized to environmental issues. Using as an example of a technical specification model, a particular case study is applied as a reference for the elaboration of other specifications that address similar objectives of the massive replacement of LEDs, based on eco-efficient environmental criteria.

Model of improved technical conditions for the procedure of contracting the supply and installation of lighting LED technology for a maritime walk of the city council of Fuengirola (This model is divided into 20 sections that we define.)

### 2.2.1. Introduction

Included in the Energy Savings and Efficiency Plan that the City of Fuengirola is carrying out, and thus complying with current legislation in this area, REEIAP and specialized regulations of application, as well as the CIE and IEC (International Electrotécnical Comision) recommendations on technical requirements required for luminaires with LED outdoor lighting technology, including UNE-EN, ISO, IEC and the standards contained therein, this specification proposes the technical conditions for the replacement of existing discharge luminaires with more efficient ones with LED technology equipped with electronic ignition equipment of variable power and with the possibility of point-to-point remote management by radiofrequency as an improvement. Furthermore, a sensitivity and environmental impact analysis of the LED luminaires is offered.

### 2.2.2. Objective

This model of the Technical Specification is intended to tender the supply, replacement and installation of 1034 energy-efficient and environmentally sustainable LED luminaires, replacing 1034 existing luminaires currently equipped with lamps (HM) and electromagnetic ballast with others equipped with LEDs, including dimmable plate and source (dimmable 1–10 V and potentiometer 100 K), and which are capable of point-to-point intercommunication by radiofrequency (Figure 1).

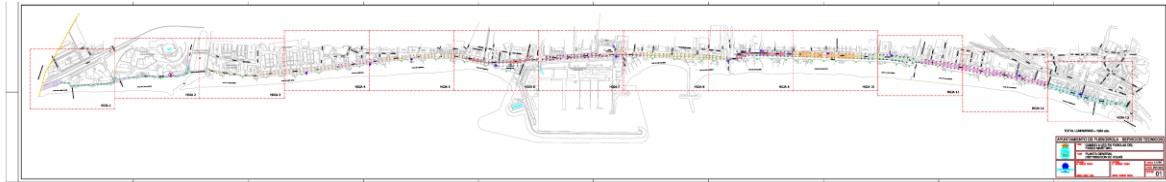

**Figure 1.** Plan of action for 6 km of Pº, Maritime, Fuengirola (Málaga). Zone of LED lighting substitutions for which we are going to apply the Public Tender Model with Environmental Criteria as a case study (Source: Own elaboration).

This document is intended to establish the technical requirements and economic, lighting, and environmental technical criteria that will be used to assess the offers which must be submitted by the competing companies in the contract award process. In the replacement of the 1034 public lighting elements, the supply of the new equipment will be included, as well as its turnkey installation (labor, auxiliary machinery, and materials), including tests, as well as the withdrawal of the current equipment to a place indicated by the City Council.

This action aims to reduce pollutant emissions and improve the efficiency and light uniformity of the Promenade and represents a significant percentage decrease in energy expenditure, in addition to

minimizing the environmental impact in all environmental categories caused by the manufacture of LED luminaires.

### 2.2.3. Characteristics and LCA Required

The lighting of the Paseo Marítimo consists of 1034 luminaries mounted on columns of 4.5 m in height and with an interdistance of about 20 m. Each column is equipped with two luminaires each, with the exception of 10 columns with one luminaire per column—see Figure 2.

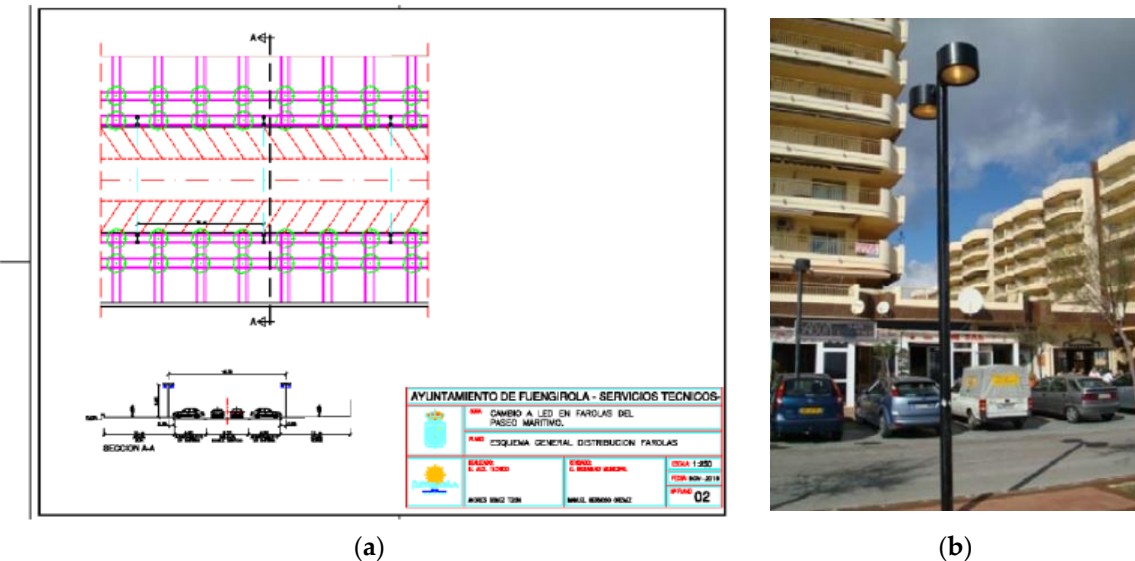

(**a**)                                                    (**b**)

**Figure 2.** Luminaire detail and their position on a plane. (**a**) Detail plan of luminaire arrangement in the street; (**b**) Luminaire detail. (Source: own elaboration).

The column is connected to a voltage of 230 VAC 50 Hz, with a single phase, and each luminaire is protected with an independent fuse located in the lower door, at 1 m above ground (Figure 3).

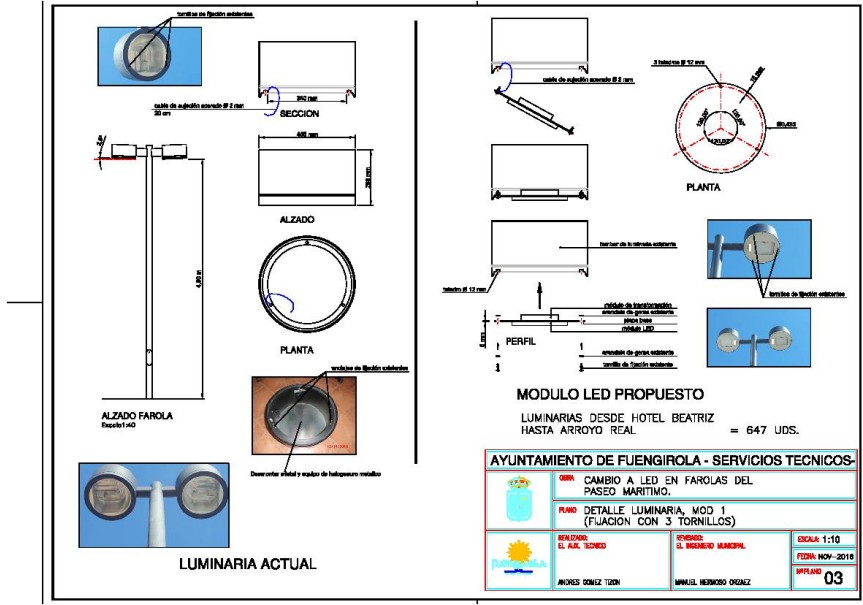

**Figure 3.** Detail of LED luminaire substitute (Source: own elaboration).

Each company participating in the contracting procedure must provide a complete lighting study carried out with a specific software DIALUX [48] program or similar to justify the adopted solution, which will specify in particular the results of the light study, with a definition of average uniformity (Um), minimum illuminance (Emín) and medium illuminance (Em).

In addition, bidders will present an environmental impact study using life cycle analysis (LCA) techniques. The environmental impact study will require a life cycle analysis using a duly justified technical and scientific methodology [49]—see Figure 4.

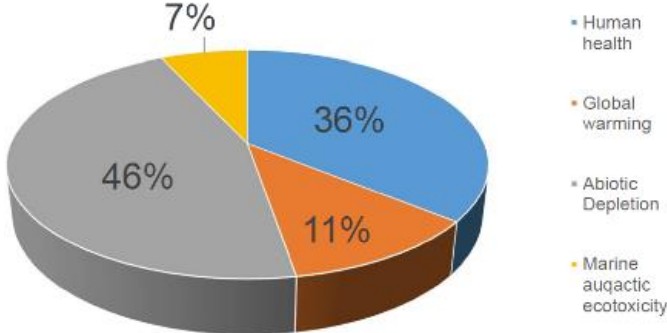

**Figure 4.** Detail of Results % of Environmental Impact Analysis (life cycle analysis (LCA)) performed for an LED Luminaire with SIMAPRO software (Method EPS 2000). Luminaire Viasol SOLIDY 50 W. (Source: Own elaboration. Dates: SOLIDY-SOLITEC LED Luminaires for LCA study).

LCA is an appropriate methodology to determine the environmental impact that occurs throughout the life cycle of products, services, or processes. It also allows for the determination of the environmental impact of any phase independently of the rest [50].

For the analysis of the LCA, which must be provided by bidding companies, any existing scientific–technical software on the market may be used, providing the technical characteristics, manufacturer, and version used—for example, SIMAPRO software (see Figure 4).

### 2.2.4. Bidding Basis Budget

The budget of the contract, before VAT, will amount to €246,278.12, with the corresponding 21% VAT amounting to the total amount of the bidding base of € 297,996.53—an amount that can be improved downwards in the offers by the bidders (Table 1).

**Table 1.** Unit and total budget (Source: own elaboration).

| N° of Luminaires (Uds) | Unit Price (€) | Total € (without V.A.T) | Total € (with V.A.T) |
| --- | --- | --- | --- |
| 1.034 | 238,18 | 246.278,12 | 297.996,53 |

This amounts to a unit budget per luminaire and point of light of €238.18, including the disassembly of existing luminaires and the assembly of the new plate equipped with an LED base plate and dimmable source (Dimmable 1-10 V and 100K Potentiometer), including in the unit price the auxiliary means of elevation, auxiliary machinery, labor, and materials—all assembled, fully finished, and tested.

Once the contract has been awarded, the price will include all expenses and taxes that are incurred as a result of the supply and the complete assembly—fully tested—including transport and auxiliary machinery.

### 2.2.5. Execution Time (Deadline) and Reception of the Installation

The total term of the delivery and installation of the material object of this contract will be 12 months, counted from the moment of the signature of the Staking Verification Document. The main contractor awardee will incur a delay for the term cited in accordance with the public sector contract law.

Within a maximum period of 1 month from the total installation of the luminaires, the formal and positive act of reception will take place. When the installation is not in a condition to be received, this will be expressly stated, and instructions will be given to the contractor to remedy the defects or proceed to correct the defects in accordance with the agreement. When there is no correction or replacement, the City Council will leave the contractor's account, being exempt from the payment obligation.

### 2.2.6. Installation Guarantee

A minimum guarantee of five years is established for the installed material, against manufacturing and/or malfunction, for any element or material of the installation that causes a total failure or a loss of illuminance greater than 5%, guaranteeing the luminous performance of the products. These guarantees will be based on a use of 4350 h/year. Regarding the failure of the power system, the drivers or power supplies must maintain their operation without alterations in their characteristics, especially in consumption and start-up peaks, during the warranty coverage period.

Mechanical defects due to material or manufacturing failures will also be guaranteed. All warranty terms must be agreed upon between the winner and the manufacturer, considering it necessary that all aspects and components affected by it are reflected and included in the guarantee document. During the warranty period, the shipping and return of the damaged material will be borne by the winning company.

During the guarantee period (minimum five years), the winner will keep a deposit of 5% of the material installed at no charge or cost to the City Council in the General Warehouse of Fuengirola. This material will be the property of the winning company until the end of the guarantee of the installed product—on which date, it can be withdrawn. This is intended to guarantee maintenance work, with the material in deposit provided by the Contractor company, thus facilitating the replacement of the damaged material by the deposit, without causing damage to the citizen and allowing a faster and more efficient replacement of defective materials. The removal of defective material remains the responsibility of the winning company, as well as its subsequent replacement, repair and shipping, at no cost to this administration. In this way, we will always maintain 5% as a refueling guarantee.

### 2.2.7. Applicable Legislation

All products subject to this contract will be subject to the CE marking, which indicates that any element or component that exhibits this marking complies with the following legislation and any other associated legislation which is applicable at any time.

Bidders will be required to present the documentation and certificates necessary to justify compliance with current legislation, especially in environmental and recycling matters related to the luminaire model proposed by each of them for its supply and replacing the existing fittings—the lack of any of these documents will result in their exclusion from the award procedure.

Thus, bidders are required to present a certificate issued by a laboratory accredited by the ENAC (National Accreditation Entity) or similar international that proves that the company and all its manufacturing processes for the activity being contracted (equipment supplied) are certified by ISO 9001-2000, as well as a certificate of legislative compliance.

### 2.2.8. Applicable Regulations

Bidders will also be required to submit the necessary documentation and certificates to justify compliance with current regulations regarding the luminaire model proposed by each of them for their supply and to replace the existing fittings. The lack of any of these documents will determine their exclusion from the award procedure (Tables 2–5)

**Table 2.** Security Requirements UNE-EN standard. (Source: ISO-UNE standard and own elaboration).

| Security Requirements Standardization UNE-EN |
|---|
| UNE EN 60598-1 Luminaires. General requirements and tests.<br>UNE EN 60598-2-3 Luminaires. Particular requirements. Street lighting luminaires.<br>UNE EN 62504: 2015 General lighting. Products of electroluminescent diodes (LED) and related equipment. Terms and definitions. |

**Table 3.** Electromagnetic Compatibility. (Source: ISO-UNE standard and own elaboration).

| Electromagnetic Compatibility Standardization UNE-EN. |
|---|
| UNE-EN 61000-3-2. Electromagnetic Compatibility (EMC). Part 3-2: Limits. Limits for harmonic current emissions (equipment with 16A input current per phase),<br>UNE-EN 61000-3-3. Electromagnetic Compatibility (EMC). Part 3: Limits. Section 3: Limitation of voltage variations, voltage fluctuations and flicker in public low voltage supply networks for equipment with 16A input current per phase and not subject to a conditional connection.<br>UNE-EN 61547. General lighting equipment. EMC immunity requirements.<br>UNE-EN 55015. Limits and measurement methods of the characteristics related to radioelectric disturbance of lighting equipment and the like. |

**Table 4.** Luminaire Components. (Source: ISO-UNE standard, EIC (International Electrotechnical Commission) and own elaboration).

| Luminaire Components Standardization UNE-EN. and IEC |
|---|
| UNE-EN 62031. LED modules for general lighting. Security Requirements<br>UNE-EN 61347-2-13. Lamp control devices. Parts 2–13: Particular requirements for electronic control devices powered by direct current or alternating current for LED modules.<br>UNE-EN 62384. Electronic control devices powered by direct current or alternating current for LED modules. Performance Requirements<br>IEC 62717: 2014. LED modules for general lighting. Performance Requirements<br>IEC 62722-1: 2014. Operating characteristics of luminaires. Part 1: General requirements.<br>IEC 62722-2-1: 2014. Operating characteristics of luminaires. Part 2: Particular requirements for LED luminaires. |

**Table 5.** Luminaire Components (Source: ISO-UNE standard, CEI (Spanish Committee of Illumination) and own elaboration).

| Luminaire Components Standardization UNE-EN CIE |
|---|
| UNE-EN 13032-1: 2006. Light and lighting. Measurement and presentation of photometric data of lamps and luminaires. Part 1: Measurement and file format.<br>UNE-EN 13032-4. Light and lighting. Measurement and presentation of photometric data. Part 4: LED lamps, modules and LED luminaires.<br>CIE S025/E: 2015. Test method for LED lamps, luminaires and LED modules.<br>CIE 127-2007 LED measurement |

Both standards, IEC 62722-1 and IEC 62722-2-1 [51], are of great importance because they require the classification of luminaires according to IRC, color dispersion, flow maintenance and their efficiency in lm/W.

### 2.2.9. Other Documents to Contribute

Additionally, bidders must provide a data sheet of the luminaires indicating all the technical characteristics regarding the type of light source, power supply, optical system, materials and finishes, operating temperatures, maintenance characteristics, degree of protection, electrical characteristics (power factor according to flow and current boot), and installation features—an official datasheet of the manufacturer or distributor of the light source used in the luminaires, indicating the exact type of source used in the luminaire, as well as all the technical characteristics of the type of light source

(nominal flow at 25 °C, lumens/watt, color temperature and color performance); a certificate issued by the manufacturer or distributor of the luminaire, including the duration of the warranty and the life of the luminaire (set of light source + power supply), and the conditions that will govern the warranty in addition to the references of the font types used; a guarantee equivalent to the useful life for labor and spare parts; a certificate that includes the test and photometric study of the luminaires in accordance with the provisions of the UNE-EN 13,032 Standard (the study must provide complete data of the photometric curves in a format compatible with free DIALUX luminaire software [48], light efficiency and its performance, the color temperature and the color performance of the light source); a photometric calculation carried out by means of a DIALUX calculation program or similar and for the type arrangement described in this Specification, in which compliance with the levels of average illuminance, average uniformity, maintenance factor, etc. is justified for the luminaire proposed by each bidding company; a certificate from the manufacturer, distributor or installer which is registered in a GIS (Integrated Waste Management System), and an environmental impact study, which includes the life cycle analysis (LCA) of the materials, products, and services included in the offer. This will be carried out by scientifically and technically competent personnel and will include, as an annex, the environmental product declaration, (EPD), consisting of standardized reports that provide quantified and verifiable information on the environmental performance of the entire life cycle of the materials [49].

### 2.2.10. Initial Bases of the Project

Before starting the work, according to the contract direction, the performance of the project will be planned so that the lighting of the area is not altered, avoiding dark areas at all times and without affecting the lighting that is in service. The existing material which will not be reused will be dismantled according to the established plan and delivered to a place indicated by the City Council. All actions undertaken will be in accordance with what is specified in the different chapters of the Bid.

### 2.2.11. Previous Visit of the Installation by the Bidders

Before the presentation of their offer, bidders may visit and investigate the installation object of this tender for the purpose of formulating an offer. They must coordinate with the person in charge of the contract and will not be able, in the case of being awarded the tender, to use differences observed from the data provided in this bidding document in order to claim a contractual modification or price change.

In order to visit the installation or see the disassembled equipment prior to the tender, the bidders will need to contact the person in charge of the city council contract via email.

### 2.2.12. Samples

The presentation of a complete sample will be mandatory for the purposes of the appropriate verification, measurement and analysis of its characteristics and functionality. The samples will be delivered as an integral part of the technical offer. There must be a label clearly indicating the name or business name of the tenderer and the code and name of the article.

The Administration may carry out the tests that it deems necessary to verify the quality of the items and their adequacy for the intended supply. These tests may be of a destructive nature, and thus the delivered samples will not be returned. The samples presented together with the offers that are selected as awardees will not be returned, nor will they be considered, in any way, as a partial delivery of the awarded contract; they will remain the property of the City Council.

Failure to comply with the requirements indicated in this specification, as well as the non-suitability of the samples, will result in the exclusion of the bidder. Non-conformities or defects detected in the submitted samples, which do not imply the rejection of the offer, will be communicated to the contractor to be corrected.

### 2.2.13. Technical Characteristics

The technical characteristics are described in Table 6.

**Table 6.** Table of technical characteristics (Source: IEC, CEI and ISO-UNE Recommendations and own elaboration). THD: total harmonic distortion.

| Manufacturing Materials (in Contact with the Outside) | Requirements |
|---|---|
| Nominal Power | Minimum 70W. |
| Power Supply | Externally dimmable: 0–10V and 100K potentiometer. |
| Efficiency (lm/W) | Minimum 80 lm/W |
| Color Temperature (°K) | Minimum 3.900 y Maximum 4.500 °K |
| Estimated Useful Life | Minimum 50,000 h. |
| CRI (Chromatic Reproduction Index) | Minimum 70 |
| Optical Group Hermeticity Degree | Minimum IP 65. |
| Degree of Protection Before Impacts | Minimum IK 10. |
| Power Voltage and Frequency | AC230V/50Hz |
| Power Factor (cos $\varphi$) | Minimum 0.95 |
| THDmax current harmonics | Maximum 20% |
| THD harmonics of odd current order | Maximum 20% |
| Environmental impact LCA Analysis in all impact categories | Maximum allowed according to current legislation Impact Categories: Human Health/Ecosystem Quality/Resources/Other impact categories |

### 2.2.14. Functionality

To ensure adequate protection against saltpeter, dust and moisture penetration, the optical system, temperature control, power supply, mounting and disassembly screws of the plates containing the light source and the cooling system will be IP65 according to UNE 60598.

The luminaire will have a mountable and detachable electrical connection system that facilitates installation and maintenance. This system will not compromise the degree of tightness, electrical insulation or damage the electrical installation cable. The LED luminaires will be robust with a degree of impact resistance of IK10. The power supply must be replaced by mounting and disassembly screws. The system power factor must be greater than or equal to 0.95. The lifetime of the light source assembly shall be at least 50,000 h. The luminaires will be prepared to be managed by remote management systems; for this, the power supply will be adjustable (dimmable) with a voltage from 0 to 10 V and a 100 K potentiometer. Those luminaires which exceed the minimum power of 70 W in the improvements (LED board + source) must be placed externally, with fire resistance calculated to reduce the power to 70 W.

### 2.2.15. Photometry

The luminous efficiency of the assembly (light source + power supply) must be equal to or greater than 90 lm/W.

For the purpose of the calculation for the presentation of the Light Study, a model street will be used as a model for the evaluation of the light efficiency of the project, with a type arrangement with a distribution of luminaires at a symmetrical bilateral 20 m interdistance, a point of light height of 4.5 m, a width of the main road or roadway of 6 m, a width of double parking of 4.5 m on both sides of the main road or roadway, and the widths of the two sidewalks will be 8 m each—see the attached plan detail (Figure 2). All calculations must comply with Royal Decree 1890/2008 of November 14, REEIAE [44], which approves the regulation of energy efficiency in outdoor lighting installations and its complementary technical instructions EA-01 to EA-07, or it will be necessary to justify, that once the regulations are met, it will be possible for the projected equipment to meet the required levels.

During the installation period, the City Council may choose a sample that will be sent to a laboratory with ENAC accreditation, where the photometric values, light efficiency, performance, color temperature, color performance of the light source, the CE marking and degree of glare according to UNE-EN 13,032 will be determined [52].

### 2.2.16. Security

No elements or metal component parts may accidentally detach due to vibrations or shocks and, in case of detachment, they must not fall on public roads; this is done to avoid the possibility of accidents. The closing of the luminaires will involve high security, i.e., the lights cannot, as a result of vibrations, have an accidental opening. However, this should not be an obstacle to the maintenance service. The plate that supports the luminaire and the power supply will be attached to the outer casing by means of a folding mechanism or a clamping cable. The electronic LED body and the power supply will be attached to the circular plate by means of screws in order to facilitate maintenance work.

### 2.2.17. LCA analysis: Proposed Methodology to Evaluate the Environmental Impact as a Criterion

The life cycle analysis will be carried out using ISO 14,040 standards (Table 7) to define the principles and framework and in accordance with ISO 14,044 to describe the different stages of the analysis [53,54].

**Table 7.** Table of technical characteristics (Source: ISO standard recommendations and own elaboration).

| Standard | Description | Edition |
|---|---|---|
| ISO 14040:2006 | Environmental management. Life Cycle Assessment. Principles and framework. | 2006 |
| ISO 14044:2006 | Environmental management. Life Cycle Assessment. Requirements and Guidelines. | 2006 |
| ISO/TR 14047:2012 | Environmental management. Life Cycle Assessment. Illustrative examples on how to apply ISO 14,044 to impact assessment situations. | 2006 |

The proposed system will analyze the raw materials and energy used in the different manufacturing processes of the luminaire (anodized aluminum, plastics, polycarbonate, glass, etc.) in the laboratory and take into account the energy consumed in the production (extruded, welded, molding, machining, painting, sealing, etc.). To overcome the potential limitations, the initial assumptions are defined as follows:

- The determination of the electricity used assumes that the production mix corresponds to the Spanish energy production system.
- The cleaning of the different devices used in the process is discarded as it does not contribute a considerable percentage of the total energy.
- The transport of material has been considered, for each component of the material, to be the relevant distance covered from the quarry processing or extraction point to the study laboratory for calculation purposes [55].

The evaluation of the impact of the life cycle for the construction of LED luminaires will be carried out using the LCA SimaPro 8.30 software [56], or another similar program that is widely used in the technical or scientific field and which has sufficient guarantees for its use [57].

For the realization of the LCA, any existing software on the market may be used, providing the technical characteristics, manufacturer, version used, and guarantees of scientific use with proven results are reviewed and scientifically verified.

2.2.18. Evaluation of Environmental impacts in the Different Categories

In general, the results of the LCA performed with SimaPro 8.30 software [56] or similar software must contain separate results in the four main categories of damage-oriented impact: human health, ecosystem quality, climate change, and resources.

We intend to incorporate the LCAanalysis into the specifications as an environmental criterion and compare the results obtained through a sensitivity analysis of the results, contrasted by at least two methods of impact assessment.

The use of at least two contrast methods is proposed for Sensitivity Analysis. Proposed methods include the IMPACT 2002+ Method [58,59] and the ReCiPe Endpoint v1.12, [60], EPS2000 method and CML IA-baseline method, or any other two methods that offer similar contrast guarantees. On the one hand, the IMPACT 2002+ method [58,59] considers the four categories of damage-oriented impact—human health, ecosystem quality, climate change and resources—separately. On the other hand, the ReCiPe Endpoint v1.12 method [61] considers only three categories of damage-oriented impact: human health, ecosystem quality and resources. In contrast, the EPS 2000 method and CML IA-baseline method provide us with information concerning the quantity and importance of $CO_2$ emissions to the atmosphere.

Any other method for contrast may be used which offers sufficient guarantees and is scientifically solvent and technically proven.

With the data provided previously, we can perform an evaluation of the environmental impact of the samples using SimaPro 8.30 software. Subsequently, we perform a Sensitivity Analysis [62] comparatively by at least two evaluation methods to verify possible differences in the results.

Tables 8–10 show the indicators of impacts on different categories of some of the proposed methods for the sensitivity impact analysis.

**Table 8.** Indicators of impacts according to ReCiPe Endpoint v1.12.

| Impact Category | Category Indicator | Measurement Units |
| --- | --- | --- |
| Quality of the ecosystem | FDP * | $FDP/m^2 \times year$ |
| Human health | DALY ** | People/year |
| Natural resources | Damage to resources | MJ/Kg |
| Abiotic resources *** | Exhaustion | Kg |

* Fraction of potential disappearance of the ecosystem per $m^2$ and year. ** Disability-adjusted life year: Reduction of years of life per person/year. *** Climatic, geological and geographical resources. Biodiversity.

**Table 9.** Indicators of impacts according to the EPS 2000 method.

| Impact Categories | Unit |
| --- | --- |
| Ecosystem Production Capacity | $PDFm^2yr$ |
| Human Health | Persona/yr |
| Damage Recourses | MJ/Kg |
| Biodiversity depletion | $PDFm^2yr$ |

**Table 10.** Indicators of impacts according to the CML 2001-IA baseline method.

| Impact categories | Unit |
| --- | --- |
| Global warming/climate change | Kg $CO_2$ equiv |
| Ozone Depletion | Kg CFC-11 equiv |
| Water acidification | Kg $SO_2$ equiv |
| Creation of photochemical oxidant | Kg $C_2H_4$ equiv |
| Water eutrophication | Kg $PO_4$ equiv |

2.2.19. Description of the Possible Improvements

Within the Specification of Technical Requirements that governs the procedure for contracting the supply and installation of LED technology luminaires and the support for improving the lighting of the Paseo Marítimo de Fuengirola (Málaga), it is considered appropriate to propose the following improvements to protect luminaires against high voltage or high voltage peaks:

- Improvement 1: Cirprotec NSB-10/230-C3-DD or similar surge protection equipment should be included in each luminaire (Table 11).
- Improvement 2: The installation of 13 control panels and lighting protection in the Pº Maritime (see drawings), as well as 13 pieces of temporary and permanent surge protection equipment with automatic reconnection V - Check 4RC or similar (Table 12).

**Table 11.** Improvement 1: Protection of luminaires.

| Equipment NSB-10/230-C3-DD. (Uds) | Unit Price € (Including Labor and Materials, Fully Installed and Tested) | Total € (net) |
|---|---|---|
| 1.034 | 20.16 | 20,845.44 |

**Table 12.** Improvement 2: Protection of electrical control panels.

| of Equipment. V-Check 4RC. (Uds) | Unit Price € (Including Labor and Materials, Fully Installed and Tested) | Total € (net) |
|---|---|---|
| 13 | 280.05 | 3,640.65 |

2.2.20. Quantifiable Valuation Criteria, Automatically Including Environmental Awards

The following will be an essential requirement for the evaluation of the offers:

(a) Compliance with the minimum technical requirements required in this technical specification.

(b) The delivery of the required documentation in this technical specification.

(c) The presentation of the sample: this sample, at the time of delivery, will be evaluated, and under no circumstances may it be replaced by any other item with different characteristics than the one presented with the offer.

(d) A justification of the environmental impact analysis must be presented in the different impact categories performed as LCA. The LCA environmental impact study will require an analysis of the life cycle using LCA SimaPro 8.30 software [56], or any other similar program that is widely used in the technical or scientific field with sufficient guarantees for its use [55]. Subsequently, a sensitivity analysis will be performed using the ReCiPe Endpoint v1.12 characterization method and/or the Impact 2002+ method, EPS2000 method and CML IA-baseline method, or any other two methods which have been duly technically and scientifically justified.

*2.3. Applied Methodology*

The evaluation of the proposals will be carried out in accordance with the documentation presented, regulated based on a total score of 100 points, according to the criteria listed in Table 13.

**Table 13.** Improvement 2: Protection of electrical control panels (Source: own elaboration).

| Damaged Categories | Weight (%) |
|---|---|
| 1. Global economic valuation | 20 |
| 2. Economic evaluation of the best improvement | 20 |
| 3. Lighting valuation | 20 |
| 4. Technical assessment of samples | 20 |
| 5. Assessment of environmental impacts | 20 |

Once the different criteria have been defined, we will apply the PROMETHEE Method [43,63] to analyze and decide on the best of the proposals presented including the environmental criteria [37].

The PROMETHEE method is one of the most recent methods in this field. The first theoretical references to the method are from the mid-1980s, referencing its creator Jean-Pierre Brans—a professor at the University of Québec (Canada) [64]. It is based on the use of normal and pseudo-criteria of different types (linear, ladder, Gaussian) to establish the overarching relationships between two alternatives. The prescriptive approach, called Promethee, provides the decision maker with a complete and partial classification of actions. Promethee's applications to complex multi-criteria decision scenarios have produced extensive results in problems involving planning, resource allocation, priority setting and selection among alternatives. Other areas of its application have included forecasts, talent selection and tender analysis. The PROMETHEE method attempts to "model" a very complex structure of the problem, and by engaging a team of experts, it makes the evaluation procedure and the weight determination of each group of criteria which are evidently in conflict as objective as possible. The influence exerted by the parameters of each criterion's weight on the obtained solution was estimated by different scenarios of weight determination and by an analysis of the stability of criterion weights [65].

Using the different criteria mentioned, their weights were calculated and used in the PROMETHEE method (Visual Promethee 1.4 academic edition documentation, 2019) [66] and a multi-criteria decision tool was used to rank the 10 proposals presented in the public tender as will be explained in point 3 (Results). In general, the PROMETHEE method was designed to deal with multi-criteria problems and their associated evaluation tables. The additional information needed to run PROMETHEE is quite clear and understandable by both the analysts and the decision-makers. It consists of information regarding the criteria and information within each criterion. The weights represent the relative importance within the criteria—these weights are non-negative numbers which are independent of the measurement units of the criteria [63]. The higher the weight, the more important the criterion. In this research, all the weights for the energy performance indicators were equal and had the same influence on the ranking [37,43].

Multi-criteria analysis (PROMETHEE Method) can adequatel +y address a variety of sustainable development dilemmas in decision-making, especially when applied to complex project evaluations involving multiple objectives and multiple stakeholder groups. Such evaluations are typically geared towards simultaneously satisfying private economic goals, broader social objectives and environmental targets. We show that, under specific conditions, a variety of stakeholder-oriented approaches may be able to contribute substantively to the resolution or improved governance of societal conflicts and the pursuit of the public good in the form of sustainable development [67,68].

To carry out this process, once the different criteria and the assigned weights have been defined, the preference indices between each two alternatives are calculated. The preference index (cij) between two alternatives, (Ai) and (Aj,), is the sum of the weights of the criteria in which the alternative (Ai) exceeds (Aj,) (Tables 15 and 16 for our case study).

$$C_{ik} = \sum_j w_j S_j\left(a_i, a_j\right) \quad S_j\left(a_i, a_j\right) \text{ value "1" si } a_i, > a_j \tag{1}$$

Subsequently, the incoming $\phi_i^{+\prime}$ (positives flows) and outgoing $\phi_i^{-}$ (negative flows) flows of each alternative are calculated as the sum, respectively, of their preference indices over the others and of the others over it (Tables 17 and 18).

$$\phi_i^+ = \sum_k c_{ik} \quad \phi_i^- = \sum_k c_{ki} \tag{2}$$

Incoming flows express the extent to which one alternative dominates over the others, while outgoing flows express the extent to which this alternative is dominated by the others.

Finally, using the flows, each alternative is prioritized over the others, as follows:

A$_i$ exceeds A$_k$, if any of the following cases occur:

$$\left[\phi_i^+ \; > \; \phi_k^+ \; \text{and} \; \phi_i^- \; < \; \phi_k^-\right] \tag{3}$$

$$\left[\phi_i^+ \; > \; \phi_k^+ \; \text{and} \; \phi_i^- \; = \; \phi_k^-\right] \tag{4}$$

$$\left[\phi_i^+ \; = \; \phi_k^+ \; \text{and} \; \phi_i^- \; < \; \phi_k^-\right] \tag{5}$$

See Figure 5 for an example or case study.

1.　Global economic valuation: Points from 0 to 20 are awarded. The best economic offer will be valued with a maximum of 20 points.

The lowest price will be the one that presents the minimum economic offer and will be the one that obtains the highest score. The rest of the offers will be valued according to the following formula. The lowest bid with respect to the type of tender will get the highest score.

The score of each offer will be found by a simple rule of three according to the following formula: (6)

$$Px \; = \; \frac{P \; x \; (TL - Ox)}{(TL - Om)} \tag{6}$$

where Px is the offer score, TL the type of tender, Ox the offer that is scored, Om the lowest offer, and PM, the maximum score.

The procedures provided for by article 85 of the RGLCAP [69] will be applied to assess and qualify disproportionate or reckless offers.

2.　Economic evaluation of the best improvement: Points from 0 to 20 are awarded.

The improvements will be evaluated based on their technical, functional and economic value in accordance with this municipal technical specification. The improvements must be offered in full and for their total value. Thus, if any company does not complete the offer, it will be automatically rejected.

- Improvement 1: Including, in each luminaire, Cirprotec NSB-10/230-C3-DD or similar surge protection equipment: valued at 10 points.
- Improvement 2: Installation in 13 control panels of lighting protection which exists in the Pº Maritime (see drawings) and 13 pieces of temporary and permanent surge protection equipment with automatic reconnection V-Check 4RC or similar: Valued at 10 points.

3.　Lighting valuation (Power and Efficiency): Points from 0 to 20 are awarded.
- Luminous efficiency of the equipment (Lm/W): 10 points.

The highest score in this section will be awarded to the team that presents the best light efficiency or best ratio of lumens per watt (lm/W). The calculation of the points for this section that are assigned will be based on the following formula (taking into account that, within the formula, "X" represents the value of the light efficiency offered by the bidding company and "Y" represents the value of the highest of the light efficiencies presented by the competing companies):

$$\left(\frac{X}{Y}\right) * 10 \tag{7}$$

- Lighting study with DIALUX software or similar: 10 points.

Companies that present a light study will be awarded 10 points. Those who do not submit it will be given 0 points in this section.

4.　Technical assessment of samples: Points from 0 to 20 are awarded.

The improvement of technical criteria up to a maximum of 20 points will be assessed, according to the attached distribution.

These characteristics will be measured by the City Council technicians and a CIRCUTOR AR6 or similar network analyzer, and Kyoritsu 2413R or similar clamp meters will be used.

Each technical characteristic will be assessed individually, according to Table 14 below, with the highest score given to the offer with the best technical value and with the points distributed in a linear manner to the rest of the offers.

**Table 14.** Quantified valuation of the technical characteristics of the luminaire.

| Technical characteristics | Punctuation |
|---|---|
| Starter peak (cold at 24 h) | 4 |
| Harmonic distortion | 4 |
| Type of material | 4 |
| Cooling system or justification of thermal dispersion | 4 |
| Luminaire adaptation plate anchoring system | 4 |

5.　Value of environmental impacts: Points from 0 to 20 are awarded.

The degree of environmental impact of the luminaires will be assessed in different impact categories during the manufacturing process, including transportation. The bidding companies will have to provide a study or scientific–technical report signed by a technical manager or expert in Life Cycle Analysis (LCA), with a justification for the Environmental Impact Analysis of the luminaire offered. The LCAstudy of environmental impact will require an analysis of the life cycle of the proposed LED luminaire, using Lima SimaPro 8.30 software [56], or any other similar program that is widely used in the technical or scientific field, providing sufficient guarantees for its use. Subsequently, an Impact Sensitivity Analysis will be performed on the LED luminaire proposed by the tenderer, using at least two of the following methods proposed to compare the results: the ReCiPe Endpoint v1.12 characterization method and/or the Impact 2002+ method, the EPS2000 method and CML IA-baseline method, or any other two methods that offer similar contrast guarantees and which are duly technically and scientifically justified. This is intended to contrast the results of the environmental impact obtained in a scientific way and incorporate them as an evaluation criterion.

## 3. Results

We will apply the practical results obtained in a tender for tenders (EI) to the proposed case study. We will analyze the results of the impact of the environmental criteria on the results of the Contest by applying, as a Valuation Criteria Method, the multicriteria analysis carried out with the PROMETHEE Method [37,40,41,63].

In Table 15, we show the criteria to be evaluated, the weights of the criteria and the proposals presented for the 10 companies evaluated for the proposed case study.

Subsequently homogenize the matrix. (Table 16) and we apply the PROMETHEE method (Table 17) to clause the positive flows ($\phi^+$) and the negative flows ($\phi^-$) (Table 18).

**Table 15.** Matrix of criteria, weights and evaluations made to the 10 bidding companies who submitted to the public tender.

| Criteria | 1. Global Economic Valuation Euros | 2. Economic Evaluation of the Best Improvement 0–20 points | 3. Lighting Valuation 0–20 Points | 4. Technical Assessment of Samples 0–20 Points | 5. Assessment of Environmental Impacts 0–20 Points |
|---|---|---|---|---|---|
| **Weight (%)** | 20 | 20 | 20 | 20 | 20 |
| **Bidding Companies** | | | | | |
| E1 | 236,287.15 | 20 | 20 | 20 | 10 |
| E2 | 246,278.12 | 20 | 20 | 16 | 20 |
| E3 | 239,458.90 | 10 | 20 | 12 | 10 |
| E4 | 240,453.24 | 0 | 10 | 20 | 15 |
| E5 | 230,954.13 | 10 | 20 | 16 | 10 |
| E6 | 238,556.29 | 10 | 20 | 16 | 10 |
| E7 | 245,328.97 | 10 | 20 | 20 | 20 |
| E8 | 228,429.78 | 5 | 10 | 16 | 10 |
| E9 | 233,720.10 | 5 | 20 | 12 | 20 |
| E10 | 236,574.56 | 10 | 20 | 16 | 10 |

**Table 16.** Matrix of weights and criteria homogenized.

| Criteria | Global Economic Valuation | Economic Evaluation of the Best Improvement | Lighting Valuation | Technical Assessment of Samples | Assessment of Environmental Impacts |
|---|---|---|---|---|---|
| **Weight** | 0.20 | 0.20 | 0.20 | 0.20 | 0.20 |
| **Bidding Companies** | | | | | |
| E1 | 0.96 | 2.00 | 0.67 | 1.00 | 0.50 |
| E2 | 1.00 | 2.00 | 0.67 | 0.80 | 1.00 |
| E3 | 0.97 | 1.00 | 0.67 | 0.60 | 0.50 |
| E4 | 0.98 | 0.00 | 0.33 | 1.00 | 0.75 |
| E5 | 0.94 | 1.00 | 0.67 | 0.80 | 0.50 |
| E6 | 0.97 | 1.00 | 0.67 | 0.80 | 0.50 |
| E7 | 1.00 | 1.00 | 0.67 | 1.00 | 1.00 |
| E8 | 0.93 | 0.00 | 0.33 | 0.80 | 0.50 |
| E9 | 0.95 | 1.00 | 0.67 | 0.60 | 1.00 |
| E10 | 0.96 | 1.00 | 0.67 | 0.80 | 0.50 |

**Table 17.** Matrix of positive ($\phi^+$) and negative ($\phi^-$) flows.

|  | E1 | E2 | E3 | E4 | E5 | E6 | E7 | E8 | E9 | E10 | Flow (+) |
|---|---|---|---|---|---|---|---|---|---|---|---|
| E1 | 0.00 | 0.20 | 0.40 | 0.40 | 0.60 | 0.40 | 0.20 | 0.80 | 0.60 | 0.40 | **4.00** |
| E2 | 0.40 | 0.00 | 0.80 | 0.80 | 0.60 | 0.60 | 0.40 | 0.80 | 0.60 | 0.60 | **5.60** |
| E3 | 0.20 | 0.00 | 0.00 | 0.40 | 0.20 | 0.20 | 0.00 | 0.60 | 0.20 | 0.20 | **2.00** |
| E4 | 0.40 | 0.20 | 0.60 | 0.00 | 0.60 | 0.60 | 0.00 | 0.60 | 0.40 | 0.60 | **4.00** |
| E5 | 0.00 | 0.00 | 0.20 | 0.40 | 0.00 | 0.00 | 0.00 | 0.60 | 0.20 | 0.00 | **1.40** |
| E6 | 0.20 | 0.00 | 0.20 | 0.40 | 0.20 | 0.00 | 0.00 | 0.60 | 0.40 | 0.20 | **2.20** |
| E7 | 0.40 | 0.20 | 0.60 | 0.80 | 0.60 | 0.60 | 0.00 | 1.00 | 0.40 | 0.60 | **5.20** |
| E8 | 0.00 | 0.00 | 0.20 | 0.00 | 0.00 | 0.00 | 0.00 | 0.00 | 0.20 | 0.00 | **0.40** |
| E9 | 0.20 | 0.00 | 0.20 | 0.60 | 0.40 | 0.20 | 0.00 | 0.80 | 0.00 | 0.20 | **2.60** |
| E10 | 0.20 | 0.00 | 0.20 | 0.40 | 0.20 | 0.00 | 0.00 | 0.60 | 0.40 | 0.00 | **2.00** |
| **Flow (+)** | **2.00** | **0.60** | **3.40** | **4.20** | **3.40** | **2.60** | **0.60** | **6.40** | **3.40** | **2.80** | |

**Table 18.** Positive ($\phi^+$) and negative ($\phi^-$) flows.

|  | Flow (-) | Flow (+) |
|---|---|---|
| E1 | 2.00 | 4.00 |
| E2 | 0.60 | 5.60 |
| E3 | 3.40 | 2.00 |
| E4 | 4.20 | 4.00 |
| E5 | 3.40 | 1.40 |
| E6 | 2.60 | 2.20 |
| E7 | 0.60 | 5.20 |
| E8 | 6.40 | 0.40 |
| E9 | 3.40 | 2.60 |
| E10 | 2.80 | 2.00 |

Once the positive and negative flows have been calculated, we will graphically represent the results; see Figure 5.

Graphing the results of the positive and negative flows for each bidding company, we find that companies with higher positive flows and smaller negative flows will be those represented in the upper right of the graph or Figure 6. These companies are the ones that would obtain the best results according to the criteria applied. In particular, the best would be the bidding company E2, followed by companies E7 and E1, which coincidentally are the ones that have presented the best results in the environmental criteria—see Figures 5 and 6.

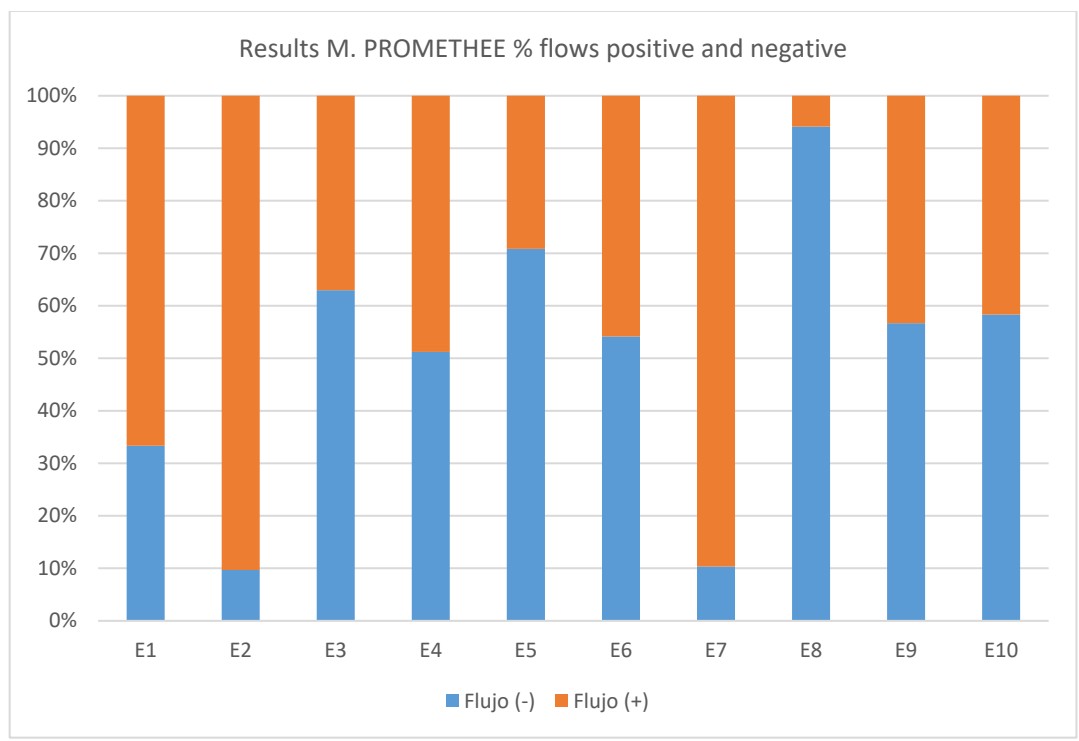

**Figure 5.** Graphical representation of the positive and negative flows for each of the bidding companies submitted to the Public Tender (E$_i$) Source: Own elaboration.

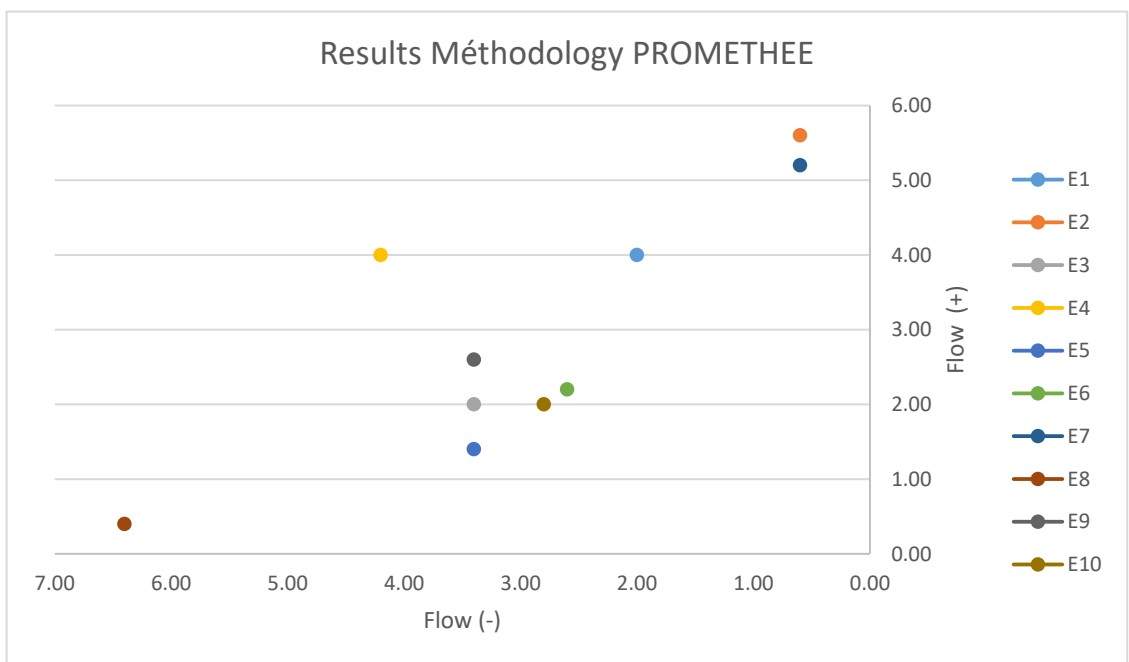

**Figure 6.** Graphical representation of the positive and negative flows for each of the bidding companies submitted to the Public Tender (E$_i$) Source: Own elaboration.

## 4. Discussion

The implementation of environmental criteria in the Public Tenders for the bidding for a change to LED technology on a large scale requires the environmental criteria to be given the sufficient importance they deserve.

LCA is shown to be an adequate solution to assess the environmental impacts incurred in the manufacture and transport of LED luminaires. Implementing the analysis of these impacts as an environmental criterion would contribute decisively to the massive substitutions with LEDs that are being carried out in the EU and around the world in the interests of environmental sustainability.

Multi-criteria analysis using the PROMETHEE Methodology has been demonstrated as a viable solution for application as an analysis method in public tender procedures, presenting itself as an effective means of valuing offers with objective criteria in terms of environmental sustainability [37,43].

Looking at the results, we see that it is not necessarily the companies that have the highest economic casualties that are best positioned for an award. In our case study, companies E7 and E2 were two of the companies that had practically not carried out an economic withdrawal; they presented an economic offer practically at the type of tender. However, they opted for LED luminaires with a very low environmental impact and so, in the end, they were two of the best placed companies when applying the PROMETHEE multicriteria analysis method.

This demonstrates that both the LCA and the environmental impact assessment criteria and the PROMETHEE methodology can be perfectly applied as a selection model in public bidding contests for massive changes to LED in our cities.

## 5. Conclusions

Energy efficiency and sustainability are strategic lines promoted by public administrations. In particular, in the field of lighting, LED technology is a reality that is increasingly implemented to illuminate our cities.

In the public administrations promoted by the EU, every day, more projects oriented to the large-scale replacement of LED discharge lamps are promoted through public competitions supported by technical specifications, which allow their legal processing and subsequent execution.

In most of these technical specifications, only technical aspects (lighting, electricity, economic or energy efficiency) associated with the characteristics of the luminaires are addressed—aspects that are valued in the form of technical criteria. In general, these competitions proceed without objectively addressing, as a selection criterion, aspects of environmental impacts generated in the manufacturing processes of LED luminaires.

In this work, we present, as a case study, a technical specification and environmental specifications model, which jointly evaluate both aspects and allow us to guarantee with greater security the proper functioning, efficiency and sustainability of the selected LED luminaires, supporting this selection in the PROMETHEE multi-criteria advanced evaluation methodology applied by different scientific studies [37,39,43,44].

This model of technical specifications for public tenders that we propose is intended as an ambitious attempt to evaluate a case of massive LED replacement. On the one hand, we evaluate economic aspects or delivery (term), and on the other hand, the technical and electrical aspects (peaks of current at startup, harmonics distortion, thermal dissipation) and lighting techniques (photometry, light efficiency, uniformity, medium and extreme lighting) are included. Furthermore, as an innovative criterion, we include the environmental impacts produced in the manufacture and transport of the chosen LED luminaire, analyzing with LCA techniques the environmental impacts incurred in the manufacture of LED luminaires.

This methodology and the technical specifications model proposed are presented as a novelty that addresses environmental criteria focused on environmental impacts. To do this, bidders must present the LCA of their LED luminaires using specialized LCA software—SimaPro 8.30 or similar—with sufficient scientific–technical guarantees.

In addition, another novel contribution consists in applying the PROMETHEE Methodology in the selection process as a model base for the elaboration of the Technical Specifications for the tendering of public lighting works and installations backed by eco-efficient environmental criteria [37,41,68,69].

On the other hand, taking into account both general sustainable development and the regulations of energy efficiency, it can be deduced that the development of new materials in the manufacture of equipment manufactured with LED, SSLL, OLED [70] or third-generation LED technology based on TADF emitters for lighting [71] should be analyzed with LCA techniques to assess their impacts, and based on the results, the decision to opt for these new LED developments in the immediate future can be supported [72].

In addition to measuring the LCA of the luminaires, we evaluate the use of by-products or waste and their recycled incorporation into the industrial life cycle [55,58]. Consequently, the products with the lowest impact will allow a reduction in $CO_2$ and $CH_4$ emissions, as well as a reduction in the consumption of energy and resources.

In this way, the use of low-cost recycled materials with a low environmental impact, which are also close to the production centers, and their return to manufacturing processes through recovery, would be encouraged by the public administrations, allowing the promotion of circular economic strategies. In this regard, it is noteworthy that tests are already being carried out regarding the use of electrical plastic waste to produce syn-gas or gasification gas [73–75] or as a material for sustainable construction [35]. It is possible to reuse the metal base, which is of great value mainly as aluminum for industrial reuse, as an electrical conductor, or as electrodes in the hydrolysis processes for the production of $H_2$ [76–78].

**Author Contributions:** Conceptualization, M.J.H.-O. and J.A.L.-M.; methodology, M.J.H.-O.; software, M.J.H.-O. and J.A.L.-M.; validation, H.-O.M.J. and P.B.; formal analysis, M.J.H.-O. and P.B.; investigation, M.J.H.-O.; resources, M.J.H.-O. and P.B.; data curation, M.J.H.-O.; writing—original draft preparation, M.J.H.-O.; writing—review and editing, M.J.H.-O.; visualization, M.J.H.-O. and R.L.-G.; supervision, P.B.; project administration, M.J.H.-O. and P.B.; funding acquisition, P.B.

**Funding:** This research received no external funding directly; however, in this work, PhD Hermoso-Orzáez Manuel Jesús participated as a main researcher in the external research at the Center Instituto Politécnico de Portalegre (Portugal) within the ECO2CIR INTERNATIONAL PROJECT-PROJECT OF CROSS-BORDER COOPERATION FOR THE INTRODUCTION OF THE ECOLOGICAL AND CIRCULAR ECONOMY THROUGH THE PREVENTION OF IMPROVEMENT OF RECYCLING, THE MANAGEMENT AND VALORIZATION OF RESIDUES IN THE REGIONS CENTRO, EXTREMADURA AND ALENTEJO.- INTERREG ESPAÑA-PORTUGAL supervised by PhD Brito Paulo, with the aim of looking for Models of Specifications that serve as a reference for the massive LED replacements that are expected to be carried out in the short and medium-term in the Alentejo (Portugal) and Extremadura (Spain) regions.

**Conflicts of Interest:** The authors declare no conflict of interest.

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
