# Peer review of "Environmental Criteria for Assessing the Competitiveness of Public Tenders with the Replacement of Large-Scale LEDs in the Outdoor Lighting of Cities as a Key Element for Sustainable Development: Case Study Applied with PROMETHEE Methodology"

_sustainability, doi:10.3390/su11215982_

Round 1

Reviewer 1 Report

This paper showed assessing competitiveness of public tenders with replacement of LEDs in cities based on the real life sustainable development. 

The manuscript is, in general, well written and organized. However, there are several mistakes or improvements to make regarding the format of the document, as commented below. 

Figure1 and 3 need better high definition image than now. Also , table formats are to be coherent eg. bold and labels, etc. 

There is a foreign language rather than English used in the subtitle of 2.3.

Conclusions should be rewritten to understand the importance of research, it is very long which makes to difficult to focus on the main topic. Also the authors should added empirical findings (Pros / cons)  of using PROMETHEE Methodology. 

Author Response

Respuesta al revisor 1:

Muchas gracias por sus comentarios constructivos. Hemos realizado una revisión completa de nuestro trabajo, modificando los aspectos indicados que hemos enumerado. Los cambios realizados se han marcado en letras rojas en el texto del artículo.

Figura 1 y 3 necesitan una mejor imagen de alta definición que ahora. Además, los formatos de tabla deben ser coherentes, por ejemplo. negrita y etiquetas, etc.

Hemos procedido a modificar las Figuras 1 y 3, dándole mayor calidad y definición, trabajando en la impresión de los planes originales del proyecto que sirvieron de base para el estudio de caso. Del mismo modo, hemos unificado las etiquetas en negrita y coherentemente con los estándares de estilo de la revista.

Hay un idioma extranjero en lugar del inglés utilizado en el subtítulo de 2.3.

Pedimos disculpas por el error. Hemos procedido a traducir el subtítulo de 2.3 al inglés

Las conclusiones deben reescribirse para comprender la importancia de la investigación, ya que es muy largo lo que hace que sea difícil concentrarse en el tema principal. Además, los autores deben agregar hallazgos empíricos (Pros / contras) del uso de la Metodología PROMETHEE.PROMETHEE Methodology.PROMETHEE Methodology.PROMETHEE Methodology.PROMETHEE Methodology.

Muchas gracias por tus interesantes comentarios. Hemos reescrito la sección Conclusiones, reduciéndola y haciéndola más corta y sintética, destacando los aspectos novedosos de nuestro trabajo.

También hemos incorporado en el texto las ventajas de usar el Método PROMETHEE, reforzándolo con bibliografía agregada para respaldar su aplicabilidad al estudio de caso analizado. Método Promethee, reforzándolo con bibliografía agregada para respaldar su aplicabilidad al estudio de caso analizado.PROMETHEE Method, reinforcing it with added bibliography to support its applicability to the case study analyzed..PROMETHEE Method, reinforcing it with added bibliography to support its applicability to the case study analyzed..

“El método PROMETHEE es uno de los métodos de superación más recientes. Sus primeras referencias teóricas datan de mediados de los años 80, siendo su creador Jean-Pierre Brans, profesor de la Universidad de Québec (Canadá) [68]. Se basa en el uso de criterios normales y seudocriterios, de diferentes tipos (lineal, escalera, gaussiano), para establecer las relaciones de superación entre dos alternativas. El enfoque prescriptivo, llamado Promethee, proporciona al tomador de decisiones una clasificación completa y parcial de las acciones. Las aplicaciones de Promethee a escenarios complejos de decisión de criterios múltiples han producido resultados extensos en problemas que involucran planificación, asignación de recursos, establecimiento de prioridades y selección entre alternativas. Otras áreas han incluido pronósticos, selección de talento y análisis de licitaciones. El método PROMETHEE, se intentó ' modelar una estructura muy compleja del problema e involucrar a un equipo de expertos para hacer lo más objetivo posible el procedimiento de evaluación y la ponderación de cada grupo de criterios que evidentemente estaban en conflicto. La influencia ejercida por los parámetros de cada peso de criterio sobre la solución obtenida se estimó mediante diferentes escenarios de ponderación de peso y mediante un análisis de la estabilidad de los pesos de criterio. [69] El método PROMETHEE es uno de los métodos de superación más recientes. Sus primeras referencias teóricas datan de mediados de los años 80, siendo su creador Jean-Pierre Brans, profesor de la Universidad de Québec (Canadá) [68]. Se basa en el uso de criterios normales y seudocriterios, de diferentes tipos (lineal, escalera, gaussiano), para establecer las relaciones de superación entre dos alternativas. El enfoque prescriptivo, llamado Promethee, proporciona al tomador de decisiones una clasificación completa y parcial de las acciones. Las aplicaciones de Promethee a escenarios complejos de decisión de criterios múltiples han producido resultados extensos en problemas que involucran planificación, asignación de recursos, establecimiento de prioridades y selección entre alternativas. Otras áreas han incluido pronósticos, selección de talento y análisis de licitaciones. Con el método PROMETHEE, se intentó 'modelar' una estructura muy compleja del problema e involucrar a un equipo de expertos para hacer lo más objetivo posible el procedimiento de evaluación y la ponderación de cada grupo de criterios que evidentemente estaban en conflicto. La influencia ejercida por los parámetros de cada criterio de peso sobre la solución obtenida se estimó mediante diferentes escenarios de ponderación de peso,PROMETHEE method is one of the most recent overcoming methods. His first theoretical references dated from the mid-80s, being its creator Jean-Pierre Brans, professor at the University of Québec (Canada) [68]. It is based on the use of normal and pseudo-criteria criteria, of different types (linear, ladder, Gaussian), to establish the overcoming relationships between two alternatives. The prescriptive approach, called Promethee, provides the decision maker with a complete and partial classification of actions. Promethee's applications to complex multi-criteria decision scenarios have produced extensive results in problems involving planning, resource allocation, priority setting and selection among alternatives. Other areas have included forecasts, talent selection and tender analysis. The PROMETHEE method, it was attempted to 'model' a very complex structure of the problem and by engaging a team of experts to make as objective as possible the evaluation procedure and the weight pondering of each group of criteria which were evidently in conflict. The influence exerted by parameters of each criterion weight on the obtained solution was estimated by different scenarios of weight pondering, and by an analysis of the stability of criterion weights.[69]PROMETHEE method is one of the most recent overcoming methods. His first theoretical references dated from the mid-80s, being its creator Jean-Pierre Brans, professor at the University of Québec (Canada) [68]. It is based on the use of normal and pseudo-criteria criteria, of different types (linear, ladder, Gaussian), to establish the overcoming relationships between two alternatives. The prescriptive approach, called Promethee, provides the decision maker with a complete and partial classification of actions. Promethee's applications to complex multi-criteria decision scenarios have produced extensive results in problems involving planning, resource allocation, priority setting and selection among alternatives. Other areas have included forecasts, talent selection and tender analysis. The PROMETHEE method, it was attempted to 'model' a very complex structure of the problem and by engaging a team of experts to make as objective as possible the evaluation procedure and the weight pondering of each group of criteria which were evidently in conflict. The influence exerted by parameters of each criterion weight on the obtained solution was estimated by different scenarios of weight pondering, and by an analysis of the stability of criterion weights.[69]

Utilizando los diferentes criterios mencionados con sus pesos, se calcularon y utilizaron en el método PROMETHEE (Visual Promethee 1.4, edición académica, documentación, 2019) [70] una herramienta de decisión de criterios múltiples utilizada para clasificar los diez (10) Ei. (Cuadro 15) propuestas presentadas en la licitación pública. En general, los métodos PROMETHEE fueron diseñados para tratar problemas de criterios múltiples y su tabla de evaluación asociada. La información adicional necesaria para ejecutar PROMETHEE es bastante clara y comprensible tanto para los analistas como para quienes toman las decisiones. Consiste en información entre los criterios e información dentro de cada criterio. Los pesos representan una importancia relativa dentro de los criterios. Estos pesos son números no negativos, independientes de las unidades de medida de los criterios [67]. Cuanto mayor es el peso, más importante es el criterio. En esta investigación, todos los pesos de los indicadores de rendimiento energético fueron iguales y tuvieron la misma influencia en la clasificación. [40] [46] diferentes criterios mencionados con sus pesos, fueron calculados y utilizados en el método PROMETHEE (Visual Promethee 1.4, edición académica, documentación, 2019) [70] una herramienta de decisión de criterios múltiples utilizada para clasificar los diez (10) Ei. (Cuadro 15) propuestas presentadas en la licitación pública. En general, los métodos PROMETHEE fueron diseñados para tratar problemas de criterios múltiples y su tabla de evaluación asociada. La información adicional necesaria para ejecutar PROMETHEE es bastante clara y comprensible tanto para los analistas como para quienes toman las decisiones. Consiste en información entre los criterios e información dentro de cada criterio. Los pesos representan una importancia relativa dentro de los criterios. Estos pesos son números no negativos, independiente de las unidades de medida de los criterios [67]. Cuanto mayor es el peso, más importante es el criterio. En esta investigación, todos los pesos de los indicadores de rendimiento energético fueron iguales y tuvieron la misma influencia en la clasificación. [40] [46]differents criterias mentioned with its weights, were calculated and used in the PROMETHEE method (Visual Promethee 1.4 academic edition documentation, 2019) [70] a multi-criteria decision tool used to rank the ten (10) Ei. (Table 15) proposals presented in the tender public. In general, the PROMETHEE methods were designed to treat multi-criteria problems and their associated evaluation table. The additional information needed to run PROMETHEE is quite clear and understandable by both the analysts and the decision-makers. It consists of information between the criteria and information within each criterion. Weights represent relative importance within the criteria. These weights are non-negative numbers, independent of the measurement units of the criteria [67]. The higher the weight, the more important the criterion. In this research all the weights for the energy performance indicators were equal and had the same influence on the ranking.[40] [46]differents criterias mentioned with its weights, were calculated and used in the PROMETHEE method (Visual Promethee 1.4 academic edition documentation, 2019) [70] a multi-criteria decision tool used to rank the ten (10) Ei. (Table 15) proposals presented in the tender public. In general, the PROMETHEE methods were designed to treat multi-criteria problems and their associated evaluation table. The additional information needed to run PROMETHEE is quite clear and understandable by both the analysts and the decision-makers. It consists of information between the criteria and information within each criterion. Weights represent relative importance within the criteria. These weights are non-negative numbers, independent of the measurement units of the criteria [67]. The higher the weight, the more important the criterion. In this research all the weights for the energy performance indicators were equal and had the same influence on the ranking.[40] [46]

El análisis de criterios múltiples (Método PROMETHEE) puede abordar adecuadamente una variedad de dilemas de desarrollo sostenible en la toma de decisiones, especialmente cuando se aplica a evaluaciones complejas de proyectos que involucran múltiples objetivos y múltiples grupos de partes interesadas. Dichas evaluaciones generalmente están orientadas a satisfacer simultáneamente objetivos económicos privados, objetivos sociales más amplios y objetivos ambientales. Demostramos que, bajo condiciones específicas, una variedad de enfoques orientados a las partes interesadas pueden contribuir de manera sustancial a la resolución o al mejor gobierno de los conflictos sociales y la búsqueda del bien público en forma de desarrollo sostenible [71]. [72] "Método PROMETHEE.) Puede abordar adecuadamente una variedad de dilemas de desarrollo sostenible en la toma de decisiones, especialmente cuando se aplica a evaluaciones complejas de proyectos que involucran múltiples objetivos y múltiples grupos de partes interesadas. Dichas evaluaciones generalmente están orientadas a satisfacer simultáneamente objetivos económicos privados, objetivos sociales más amplios y objetivos ambientales. Demostramos que, bajo condiciones específicas, una variedad de enfoques orientados a las partes interesadas pueden contribuir de manera sustancial a la resolución o al mejor gobierno de los conflictos sociales y la búsqueda del bien público en forma de desarrollo sostenible [71]. [72] “El análisis de criterios múltiples (Método PROMETHEE) puede abordar adecuadamente una variedad de dilemas de desarrollo sostenible en la toma de decisiones, especialmente cuando se aplica a evaluaciones complejas de proyectos que involucran múltiples objetivos y múltiples grupos de partes interesadas. Dichas evaluaciones generalmente están orientadas a satisfacer simultáneamente objetivos económicos privados, objetivos sociales más amplios y objetivos ambientales. Demostramos que, bajo condiciones específicas, una variedad de enfoques orientados a las partes interesadas pueden contribuir de manera sustancial a la resolución o al mejor gobierno de los conflictos sociales y la búsqueda del bien público en forma de desarrollo sostenible [71]. [72] “Método PROMETHEE.) Puede abordar adecuadamente una variedad de dilemas de desarrollo sostenible en la toma de decisiones, especialmente cuando se aplica a evaluaciones complejas de proyectos que involucran múltiples objetivos y múltiples grupos de partes interesadas. Dichas evaluaciones generalmente están orientadas a satisfacer simultáneamente objetivos económicos privados, objetivos sociales más amplios y objetivos ambientales. Mostramos que, bajo condiciones específicas, Una variedad de enfoques orientados a las partes interesadas pueden contribuir de manera sustancial a la resolución o mejorar la gobernanza de los conflictos sociales y la búsqueda del bien público en forma de desarrollo sostenible [71]. [72] "Multi-criteria analysis (PROMETHEE Method.) can adequately address a variety of sustainable development dilemmas in decision-making, especially when applied to complex project evaluations involving multiple objectives and multiple stakeholder groups. Such evaluations are typically geared towards satisfying simultaneously private economic goals, broader social objectives and environmental targets. We show that, under specific conditions, a variety of stakeholder-oriented approaches may be able to contribute substantively to the resolution or improved governance of societal conflicts and the pursuit of the public good in the form of sustainable development [71]. [72] “PROMETHEE Method.) can adequately address a variety of sustainable development dilemmas in decision-making, especially when applied to complex project evaluations involving multiple objectives and multiple stakeholder groups. Such evaluations are typically geared towards satisfying simultaneously private economic goals, broader social objectives and environmental targets. We show that, under specific conditions, a variety of stakeholder-oriented approaches may be able to contribute substantively to the resolution or improved governance of societal conflicts and the pursuit of the public good in the form of sustainable development [71]. [72] “Multi-criteria analysis (PROMETHEE Method.) can adequately address a variety of sustainable development dilemmas in decision-making, especially when applied to complex project evaluations involving multiple objectives and multiple stakeholder groups. Such evaluations are typically geared towards satisfying simultaneously private economic goals, broader social objectives and environmental targets. We show that, under specific conditions, a variety of stakeholder-oriented approaches may be able to contribute substantively to the resolution or improved governance of societal conflicts and the pursuit of the public good in the form of sustainable development [71]. [72] “PROMETHEE Method.) can adequately address a variety of sustainable development dilemmas in decision-making, especially when applied to complex project evaluations involving multiple objectives and multiple stakeholder groups. Such evaluations are typically geared towards satisfying simultaneously private economic goals, broader social objectives and environmental targets. We show that, under specific conditions, a variety of stakeholder-oriented approaches may be able to contribute substantively to the resolution or improved governance of societal conflicts and the pursuit of the public good in the form of sustainable development [71]. [72] “

Hemos realizado una revisión completa del idioma utilizando los servicios de edición Editin English del editor MDPI

Reviewer 2 Report

The authors describe an article entitled "Environmental criteria for assessing the competitiveness of public tenders with replacement of large-scale LEDs in the outdoor lighting of cities as a key element for sustainable development. Case study applied with PROMETHEE Methodology". The topic of the manuscript is interesting, and the manuscript constitutes an interesting study concerning the development of Au complexes potentially usable in OLEDs.

The work is well written. Sufficient figures are included in the manuscript for comprehension and clarity. Overall, this is a good manuscript that I recommend for publication after inclusion of minor revisions. Several improvements can be brought to the manuscript.

1) The whole document deserved a language edit as mistakes can be found throughout the whole document.

2) What about the LED lifetime. This point should be commented.

3) For my point of view, recycling of LEDs for the outdoor lighting is not sufficiently discussed, considering that ageing of such devices is relatively fast.

4) At present, a third generation of devices is under development based on TADF emitters. What about the modification of LEDs systems towards the next generation of LEDs ?

Author Response

Respuesta al revisor 2:

Muchas gracias por sus comentarios constructivos. Hemos realizado una revisión completa de nuestro trabajo, modificando los aspectos indicados que hemos enumerado. Los cambios realizados han sido marcados en letra roja en el texto del artículo. Hemos realizado una revisión completa del idioma utilizando los servicios de edición Edición de inglés del editor MDPI

Todo el documento merecía una edición de idioma, ya que se pueden encontrar errores en todo el documento.

Hemos procedido a enviar el editor de idiomas MDPI para una revisión completa del idioma.

¿Qué pasa con la vida útil del LED? Este punto debe ser comentado.

Muchas gracias por tu comentario. De hecho, la vida útil o la duración de los LED es algo que preocupa mucho. Hoy es difícil saber la duración real, ya que existen factores que pueden condicionarlo, especialmente la temperatura de trabajo externa directamente relacionada con la disipación térmica y los métodos de disipación, ya sea por ventilación forzada o por disparo directo al aire, con aletas de disparo de metal. En cualquier caso, como indica el revisor, es un aspecto muy importante a tener en cuenta. Para ello lo hemos incorporado en la Tabla 6. Tabla de características técnicas.LEDs is something that worries greatly. Today it is difficult to know the actual duration, as there are factors that can condition it, especially the external working temperatura directly related to thermal dissipation and dissipation methods, either by forced ventilation or by direct firing into the air, with metal firing fins. In any case, as the reviewer indicates, it is a very important aspect to consider. For this we have incorporated in the Table 6. (Fuente: Recomendaciones IEC, CEI e ISO-UNE y elab propio). Una estimación media mínima de la vida útil de las luminarias LED. Mínimo 50,000 h. Dependiendo de la potencia y las características técnicas de la disipación térmica. El fabricante debe justificar esta duración y el técnico municipal con la muestra proporcionada por el fabricante lo validará.IEC, CEI and ISO-UNE Recommendations and own elab). a minimum average estimate of the life required for LED luminaires. Minimum 50,000 h. Depending on the power and technical characteristics of thermal dissipation. The manufacturer must justify this duration and the municipal technician with the sample provided by the manufacturer will validate it.

Desde mi punto de vista, el reciclaje de LED para la iluminación exterior no se discute suficientemente, dado que el envejecimiento de dichos dispositivos es relativamente rápido.LEDs for the outdoor lighting is not sufficiently discussed, considering that ageing of such devices is relatively fast.

Thank you very much for your interesting comments. We have rewritten the Conclusions section, and considered the aspects related to the recycling of materials. Especially interesting can be the recycling of plastic waste that mixed with biomass by 10 to 20% can be perfectly valued energetically subjected to gasification processes in a specific reactor for the production of syn-gas. As for the metallic part (normally aluminum or anoniced aluminum), as it is a metal with great thermal dissipation power, to improve the useful life of the LED luminaire,) we also contribute in the conclusion as future research line, its reuse for conductors of electricity or its reuse to build electrodes, for hydrolysis processes, for H2 production.

“In addition to measuring the LCA of the luminaires, we evaluate the use of by-products or waste and their incorporation recycled into the industrial life cycle[59] [62]. Consequently, the products with the lowest impact will allow a reduction in CO2 and CH4 emissions mainly and a reduction in the consumption of energy and resources.

In this way, the use of low-cost recycled materials with low environmental impact, which are also close to the production centers, and their return to manufacturing processes through recovery, would be encouraged from the Public Administrations, allowing the promotion of circular economy strategies. . In this regard it is noteworthy that tests are already being carried out for the use of electrical plastic waste to produce syn-gas or gasification gas [77][78][79] or as a material for sustainable construction[38]. Being able to reuse the metal base, of greater value, mainly aluminum for industrial reuse, as electrical conductors or as electrodes in the hydrolysis processes for the production of H2 [80][81][82]“

At present, a third generation of devices is under development based on TADF emitters. What about the modification of LEDs systems towards the next generation of LEDs ?

Thank you very much again for your accurate comments. In this regard we have incorporated a clarification in this regard in the Conclusions

“On the other hand, taking into account both the general sustainable development and the regulations on energy efficiency, it can be deduced that the development of new materials in the manufacture of equipment manufactured with LED, SSLL, OLED [74] or third generation LED technology based on TADF emitters for lighting [75] , they ould be analyzed with LCA techniques to assess impacts, and based on the results, support the decision to opt for these new LED developments of the immediate future. [76]..”

Hemos realizado una revisión completa del idioma utilizando los servicios de edición Editing English del editor MDPI.
